



# A digital archive of human activity in the McMurdo Dry Valleys, Antarctica

Adrian Howkins[1], Stephen M. Chignell[2], Poppie Gullett[3], Andrew G. Fountain[4,5], Melissa Brett[4], Evelin Preciado[6]

[1]Department of History, University of Bristol, Bristol, BS8 1TH, UK
[2]Institute for Resources, Environment and Sustainability, University of British Columbia, Vancouver, British Columbia, V6T 1Z4, Canada
[3]Department of History, Colorado State University, Fort Collins, Colorado, 80523, USA
[4]Department of Geology, Portland State University, Portland, Oregon, 97207, USA
[5]Department of Geography, Portland State University, Portland, Oregon, 97207, USA
[6]Department of Ecosystem Science and Sustainability, Colorado State University, Fort Collins, Colorado, 80523, USA

*Correspondence to*: Adrian Howkins (adrian.howkins@bristol.ac.uk)

**Abstract.** Over the last half century, the McMurdo Dry Valleys (MDV) of East Antarctica have become a globally important site for scientific research and environmental monitoring. Historical data can make important contributions to current research activities and environmental management in Antarctica, but tend to be widely scattered and difficult to access. We address this need in the MDV by compiling over 5,000 historical photographs, sketches, maps, oral interviews, publications, and other archival resources into an online digital archive. The data have been digitized and georeferenced using a standardized metadata structure, which enables intuitive searches and data discovery via an online interface. The ultimate aim of the archive is to create as comprehensive as possible a record of human activity in the MDV to support ongoing research, management, and conservation efforts. This is a valuable tool for scientists seeking to understand the dynamics of change in lakes, glaciers, and other physical systems, as well as humanistic inquiry into the history of the Southern Continent. In addition to providing benchmarks for understanding change over time, the data can help target field sampling for studies working under the assumption of a pristine landscape by enabling researchers to identify the date and extent of past human activities. The full database is accessible via a web browser-based interface hosted by the McMurdo Long Term Ecological Research site: http://mcmurdohistory.lternet.edu/ and the raw data are available at the Environmental Data Initiative https://doi.org/10.6073/pasta/6744cb28a544fda827805db123d36557 (Howkins et al., 2019).



**Keywords.** digital history; environmental history; history of science; human footprint; human impact; long-term ecological research; LTER

## 1 Introduction

Antarctica's McMurdo Dry Valleys (MDV) comprise the largest ice-free area in the Southern Continent and are
among the coldest, driest places on Earth.  Since their discovery in 1903, relatively few people have visited the
MDV, and the vast majority of these visitors have been involved in scientific activity.  Historical perspectives can
make valuable contributions to current scientific research in Antarctica, including the MDV (Howkins, 2016a,
2016b).  As a result of the international and often decentralized nature of Antarctic research, historical data are
widely scattered and frequently difficult to access.  Photographs and field notebooks, for example, often remain
with individual researchers.  Universities, libraries, archives, and national Antarctic programs have different
collection policies and metadata standards, and historical data have only occasionally been digitized and made
available online.  More broadly, the future-oriented nature of much scientific research in Antarctica means that
preservation of the historical record has often been a low priority.

The aim of this project was to collect historical data related to the MDV, digitize it, and make it easily available
to researchers working in the region through an online archive.  The dispersed and varied nature of historical
documents related to the MDV has necessitated an eclectic approach to data collection.  We have concentrated
our efforts on data from the United States and New Zealand, the two countries with the most significant scientific
programs in the MDV. We have also collected data from other countries with historical interests in the region.

## 2 Data

### 2.1 Data collection

Using bibliographies of MDV publications (including grey literature) (Antarctic Division, D.S.I.R., 1985; Mead,
1978; New Zealand Antarctic Programme, 1995), recommendations from other researchers, and outreach to "Old
Antarctic Explorer" organizations, we compiled a list of individuals who have worked in the MDV.  We then
contacted these individuals with requests for historical photographs and documents related to the MDV (Fig. 1).
When we received interested replies, we collected the data using one of the following approaches.  If data
(especially photographs) were already available in digital format, we arranged for files to be sent to us
electronically.  When data were not available digitally, we either arranged for the contributors to digitize their
documents locally through commercial scanning services, or we visited the researchers in person with a scanner
(either a slide scanner or a flatbed scanner depending on the nature of the data).  The in-person visits also
facilitated oral history interviews with researchers, which we recorded, transcribed, and included in the archive
(Fig. 2).

In addition to individuals, we also worked with universities, libraries, archives, and national Antarctic programs.
Sometimes data were already easily accessible online, such as the Antarctica New Zealand's digital photograph
collection (Antarctica New Zealand, 2017).  More often, however, data were only available in non-digital





formats, in which case we followed a digitization process similar to our work with the individual researchers. The full database is accessible via a web browser-based interface hosted by the McMurdo Long Term Ecological Research site: http://mcmurdohistory.lternet.edu/ and the raw data are available at the Environmental Data Initiative https://doi.org/10.6073/pasta/6744cb28a544fda827805db123d36557 (Howkins et al., 2019).

## 2.2 Metadata structure

All data in the archive are organized according to the Dublin Core (DC), an international metadata element set intended to facilitate the discovery of electronic assets (Weibel, 1997; Weibel et al., 1998). The DC is designed
to be simple and flexible, and we have tailored its core fields for the specific purposes of the archive. This provides a consistent metadata structure among the various types of resources while maintaining interoperability with other DC databases. The full list of fields and the associated definitions for each type of resource is available in the supplementary material (Table S1).

## 2.3 Georeferencing

Although the DC includes a "coverage" field for storing geospatial information, the MDV archival data cover a wide range of spatial scales with highly variable locational certainty. We therefore chose to assign each resource a geographic "place;" these range from the scale of individual huts to entire valleys. The majority of these places are associated with a set of existing geospatial vector layers acquired from the MDV Long Term Ecological
Research site (Gardner, 2016) which we augmented to include additional features. The layer list includes:

- Camps, Stations, and Huts
- Glaciers
- Lakes and Ponds
- Streams (monitored)
- Streams (not monitored)
- Stream Gauges
- Meteorological Stations
- Dry Valleys Drilling Project sites
- Miscellaneous Human Features
- Miscellaneous Natural Features

We decided that places such as valleys or mountain ranges were too general to assign a specific geospatial layer. However, these were still assigned a searchable term in the relational database, and may be assigned a geospatial location as additional layers become available.



## 3 Online interface

We developed a web-based interface to facilitate discovery, visualization, and dissemination of the data (Fig. 3). This comprises a relational database built using Drupal, a free and open source content-management system (Buytaert, 2016). The interface provides the ability to quickly filter the database using custom searches or the DC metadata elements (e.g., date range, places, people). Clicking a photograph or other resource shows a larger version as well as the full list of DC metadata elements, which link to related resources. The user can also submit an edit to the metadata, which is intended to engage the Antarctic community and fill information gaps. We also used the geospatial vector layers as the basis for an interactive web app, which supports geospatial queries and basic geographic information system functionality.

## 4 Data availability

The historical archive is a "living" dataset which we expect to grow as future contributions are collected and digitized. At the time of this writing the database comprises more than 5,000 archival resources. These include photographs, sketches, interviews (recordings and associated transcripts), maps, sketches, bibliographic citations, and other archival data (e.g. documents) (Fig. 4). The user-friendly Drupal website is maintained by the McMurdo Dry Valleys Long Term Ecological Research (LTER) project and available at http://mcmurdohistory.lternet.edu. The raw data are available at the Environmental Data Initiative https://doi.org/10.6073/pasta/6744cb28a544fda827805db123d36557 (Howkins et al., 2019). New contributions will be uploaded and cataloged via a semi-automated system. Individuals or organizations seeking to make a contribution are encouraged to contact the corresponding author for information.

## Conclusions

Access to a geospatially referenced, historic account of human activity will be a useful resource to guide current research and environmental management in the MDV. Researchers can use these data to target investigations of the long-term environmental legacy of human activity. Scientists interested in sampling pristine landscapes can search the archive to avoid previously inhabited or impacted areas. The data can also provide benchmarks and insights into function and changes in natural systems (e.g., glacier movement, lake level rise). As research, tourism, and climate change continue to shape the Southern continent, our database and approach may also serve as a template for other regions seeking to better understand their own human-environmental histories.

**Author contributions.** AH and AGF conceived the project. AH, SMC, and PG devised the metadata schema and overarching structure of the database. AH and SMC led the design of the web map and Drupal interface,



with assistance from PG, MB, and AGF. MB and AGF led the slide scanning efforts. AH and PG led archival visits and oral history interviews, with assistance from AGF and SMC. All authors collected and digitized data for the archive. AH and SC led the writing of the manuscript and SC designed the figures. All authors provided comments and edits for subsequent drafts.

**Competing interests.** The authors declare that they have no conflicts of interest.

**Acknowledgements.** This work was supported by the United States National Science Foundation (award number 1443475). We are grateful to each of the individuals and organizations that have contributed data to the archive. We would like to thank the Morgan Library at Colorado State University, specifically Sophia Linn from the Geospatial Centroid and Mark Shelstad from Archives & Special Collections. We would also like to thank Inigo San Gil, Renée Brown, Cristina Takacs-Vesbach, and Variant Studios, Inc. for assistance with Drupal development and website design.

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

**Figure 1: Examples of photographs from the archive showing human activity in the MDV. Unless otherwise stated, copyright for each item remains with the respective contributor. Detailed metadata for each item is available on the archive website (http://mcmurdohistory.lternet.edu) and the Environmental Data Initiative (Howkins et al., 2019)**

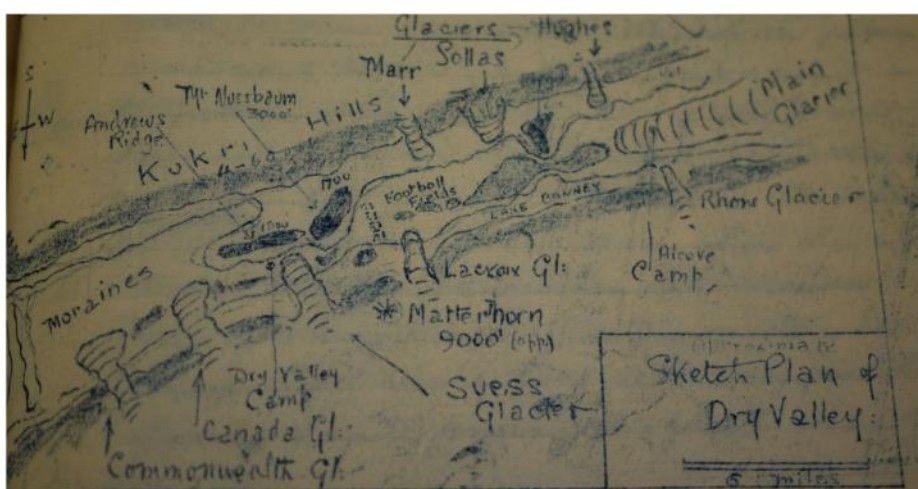

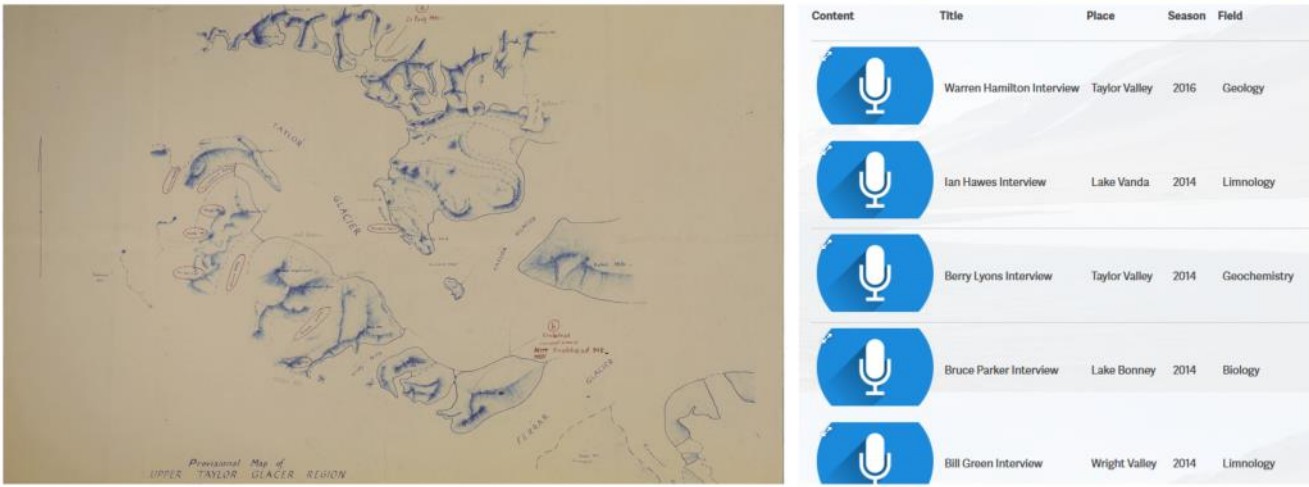

**Figure 2: Examples of non-photograph data available in the archive. Unless otherwise stated, copyright for each item remains with the respective contributor. Detailed metadata for each item is available on the archive website (http://mcmurdohistory.lternet.edu) and the Environmental Data Initiative (Howkins et al., 2019)**



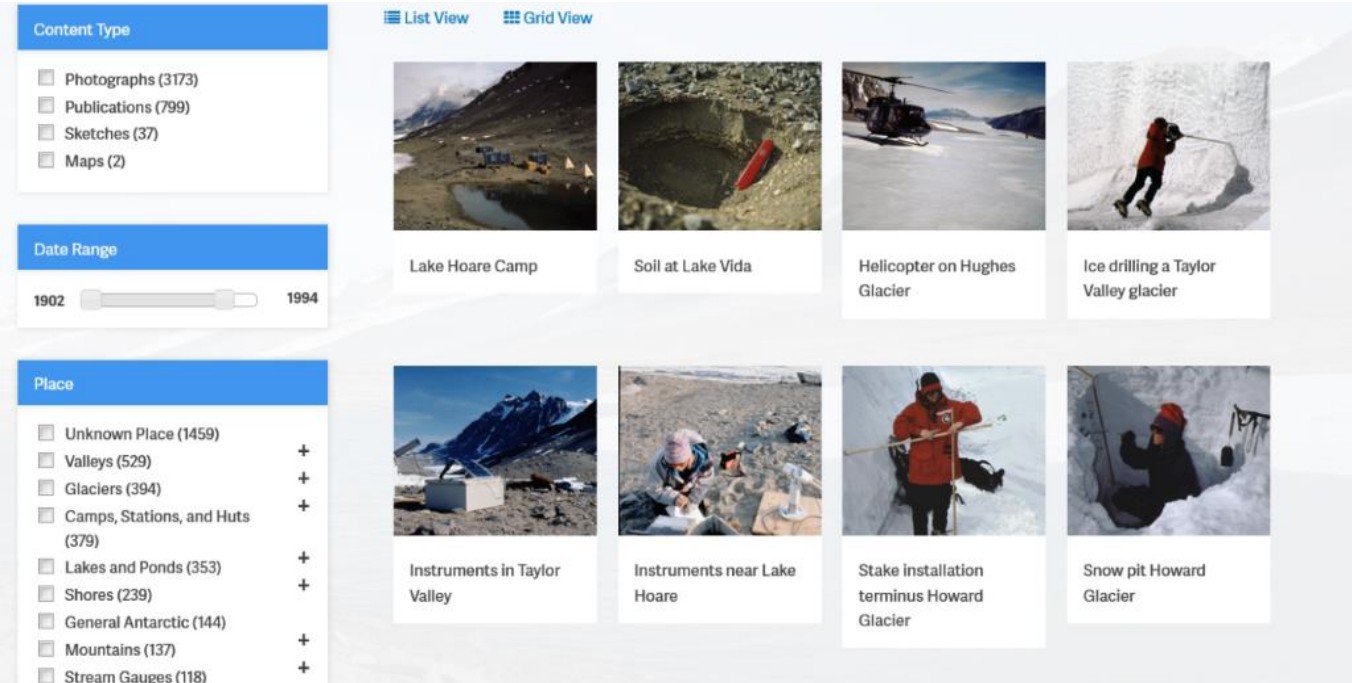

**Figure 3: Browser-based interface for searching the archive. The right-hand window refreshes as the left-hand filters are applied. Clicking on a photo or other piece of content opens a new window showing a larger image and the associated metadata.**





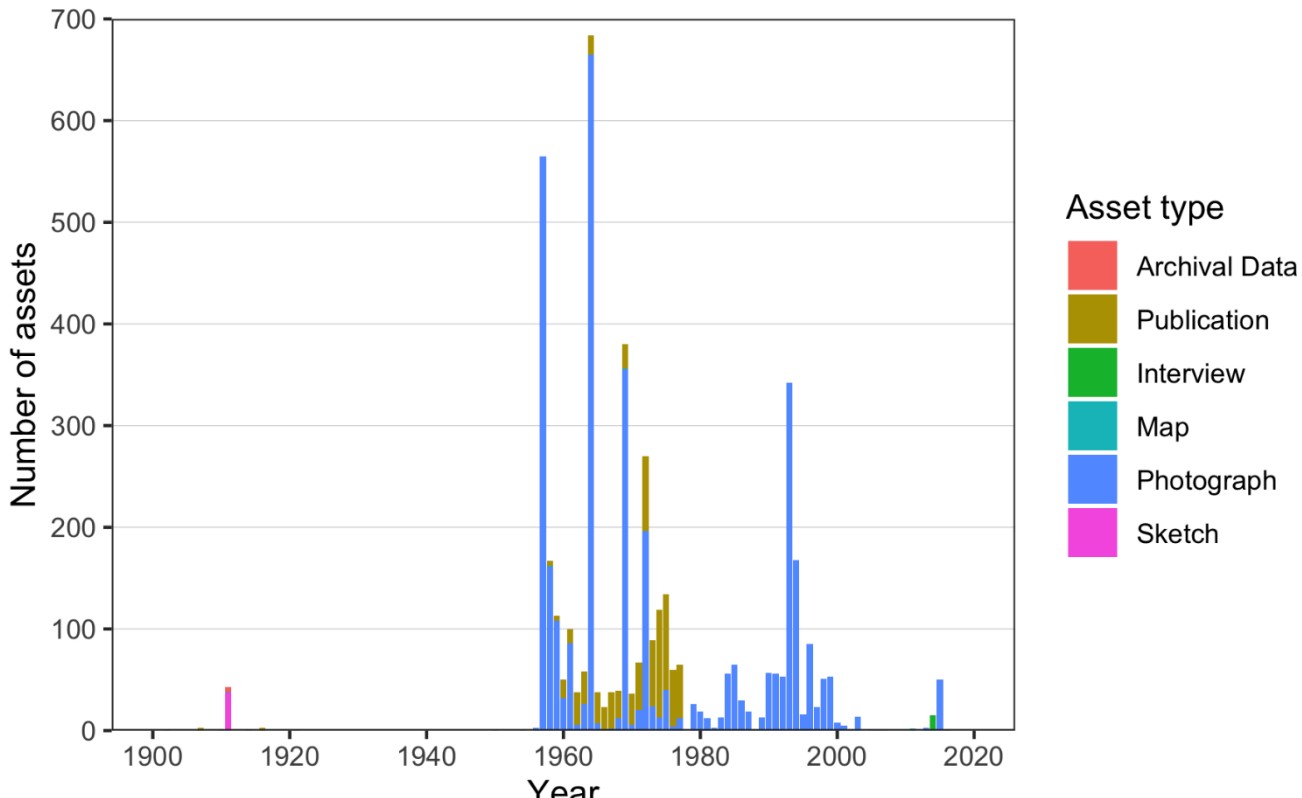

**Figure 4: Number of assets in the archive over time, based on their year of creation. Bars are colored by type of asset. Approximately 25% of the assets (1489 of 5943 total assets) were contributed without information on when they were created (i.e., "unknown season"). These assets are available in the archive but are not shown by this figure. This figure represents a snapshot at the time of writing and will be updated on the website with future additions to the archive.**