# Peer review of "A digital archive of human activity in the McMurdo Dry Valleys, Antarctica"

_Earth System Science Data, 2020_

## Referee Comment (RC1) · Warwick F. Vincent (Referee) · 7 Feb 2020

This manuscript describes an impressive collection of diverse historical material for the McMurdo Dry Valleys, an iconic site for Antarctic research by many scientists for more than 50 years. The collection includes digitized photographs, papers, interviews, maps, drawings and data sets, and is housed in the NSF-LTER website. A link is provided to a portal and search engine to discover, access and download the items, and an additional link is provided to the metadata records (in CSV format) that describe location, file type, date, source and other details for each item. As pointed out in the manuscript, this provides an excellent resource for historical analysis, environmental change assessments, land management (including tourism) and science planning, and it is a model that could be applied to other regions of the world. These historical records

are especially important given the rapid changes now taking place in the polar regions - it is a superb resource.

In response to the review criteria: Yes, the data and methods presented are new; there is high potential for the data being useful in the future; the methods and materials are described in reasonable detail, with a couple of minor suggestions (see below). The article itself is appropriate to support the publication of a data set. There was a helpful selection of material for the four figures. I could not find the supplementary material (Table S1) that is referred to.

Concerning data quality: the data set is accessible via the given identifier, and is complete as a first collection that can now be updated. The data quality is variable - some images are only poorly resolved and have many imperfections (e.g. dust on the slide, lens or scanner, poor color rendition), and the metadata are not always complete because the exact date is unknown. The authors might note this by saying that the quality and resolution are variable, but an inclusive approach was taken to maximize the scope of the database.

Further points: The images are taken by many people and "copyright for each item remains with the respective contributor" – in that case, how would permission be obtained to reproduce any of these materials, for example in another publication?

Please give the four bounding coordinates for the geographical area covered in this database.

Please explain the criteria used for selecting the publications for inclusion. They seem incomplete; for example I searched on 'Onyx River' and could only find two publications among many: 'Spatially discontinuous strain in the "semi-rigid" zone of an ice cliff. 1973. 'Principal cation concentrations for a length profile of the Onyx River, Wright Valley. 1973'

It would be useful to include the reports from the Japanese Teams in the 1970s and

1980s (Tetsuya Torii et al., as shown in Fig 1!); these include the classic paper by Yuki Yusa on the solar modeling of Lake Vanda, which is always hard to track down.

Line 29 (and twice later) states: "the raw data are available at the Environmental Data Initiative https://doi.org/10.6073/pasta/6744cb28a544fda827805db123d36557" But these are not the raw data – they are metadata (CSV files describing each data item). It seems that the actual data are the photographs, the maps, audio clips etc.

ESSD Editors may need to comment: this article will have a doi, the collection of metadata has a doi, but the data itself (the images, maps etc) are in a relational database that does not have a doi assigned - is that OK?

Story Maps: First Western Journey, 1911 Follow the route of the first expedition into the Dry Valleys in 1911. Maybe this should be 'first exploration' because the first expedition visit was in December 1903, led by Scott, who wonderfully proclaimed at that time: "It is certainly a valley of the dead."

––––––––––––––––––––––––––

---

## Referee Comment (RC2) · Anonymous Referee #2 · 24 Feb 2020

This manuscript describes an invaluable collection of diverse historical material from key US and NZ locations in the MDVs. I believe compiling these resources is long overdue, and hope this online resource will provide a foundation for cataloging all future events and research in the MDVs (by all nations working in the ASMA). I see no issues with the paper as it stands. I agree with the other reviewer, having had a look at examples of the various resources online, some are of better quality that others, and perhaps this should be mentioned. But I believe the online tool is a great resource, and could prove especially useful as we start to experience rapid environmental change in the MDVs.

This is just a comment to the authors - I wonder whether the web-browser interface might also be able to house historic US human movement data
Interactive
comment

- similar to a programme of work you probably know called the Antarctic Data Analysis project, run out of Manaaki Whenua - Landcare Research NZ, (https://www.landcareresearch.co.nz/science/soils-and-landscapes/terrestrial-data-analysis-ross-sea-region/data-analysis/human-movement) whereby human movement data (camp and sampling locations) were accessed from Antarctica New Zealand, annual data inputting into a database, and forms a layer in this management tool. Once the tool is up and running, data will be displayed and searchable, to build up a picture of historical human movement (by the NZ programme). It is a combination of these historic photos, maps, oral interviews, publications etc, and data, such as the human movement data, that will be very useful for environmental managers and decision makers as visitor (science and tourist) and environmental pressures increase in the coming years.

I liked the story maps. More of these would be great! I am sure the NZ Antarctic Programme has many many photos of the decommissioning of the Vanda Station (https://adam.antarcticanz.govt.nz/nodes/view/37814). It could be great to have all these resources combined and available on one site - but I know this is not possible, but perhaps adding links to other sites with reputable information would be useful.

The manuscript is well written. Great work on bringing together all those historic resources into one searchable tool.

---

## Author Comment (AC1) · 27 Feb 2020

We appreciate the reviewer's thoughtful comments and detailed assessment of the archive. We have provided responses to each point below.

Supplementary material: thank you for bringing this to our attention, we have attached the supplementary material (Table S1) to this comment and will include it in the final publication.

Data quality: we agree that it is important to note the variable quality of the data and metadata and will add an explanation to the revised manuscript as suggested.

Copyright: Unless otherwise stated, copyright for each item in the database remains with the respective contributor. If a reader wanted to gain permission to use the material

in another venue, they would need to contact us or the copyright holder directly. We agree that this is important to note in the manuscript, and will add it to the final version.

Bounding Coordinates: The coordinates are as follows (decimal degrees): North (-77.504), South (-77.642), East (164.319), West (161.111). We will include these in the final manuscript.

Publications: Currently, the publications included in the archive are from the three-volume bibliography of Dry Valleys publications referenced in the first paragraph of section 2.1. Together, these three volumes cover published works from 1907 through 1994. We have identified hundreds of additional publications through a Web of Science query covering the entire written record, and plan to add their references to the database in the future. Regarding publications such as those produced by the Japanese research team: these are indeed included in the archive, and the reviewers' comment brought to our attention that the backend of the database had not been configured to index the 'Bibliographic Citation' field, which is why the search was not returning all of the results. We have since adjusted the search configuration, and publications such as those by Yuki Yusa now appear in the search results (e.g., type "Yusa" in the Search bar). The dearth of results when searching for publications on the Onyx River likely reflects the term "Onyx" being seldom used in the title or abstract of earlier Dry Valleys publications. This will change once we include the additional publications identified through the Web of Science.

Raw data: It is correct that the Environmental Data Initiative only holds the metadata. It may be possible to upload the raw data as well in the future, but the reviewer is correct in stating that presently, the raw data are the photographs, maps, etc. available in the digital archive. We thank the reviewer for bringing this to our attention and will clarify this text in the final version of the manuscript.

Story Map: We have amended the text on the story maps page to: "Follow the route of the 1911 expedition into the Dry Valleys, which was the first to spend significant time
exploring the region."

Please also note the supplement to this comment:
https://www.earth-syst-sci-data-discuss.net/essd-2020-2/essd-2020-2-AC1-
supplement.zip

---

## Author Comment (AC2) · 1 Mar 2020

Thank you for your thoughful and encouraging review.

We appreciate the comparisons to the Antarctic Data Analysis project from New Zealand, and we think adding annual data on human movement is an excellent suggestion. We have considered this for the archive, but relocating historical movable field camps is very time consuming and we did not have the resources to do so in this project. Moreover, while data on 'modern' (i.e., past 10-20 years) movable camps may exist in some compiled format, the United States National Science Foundation data policy has prevented us from being allowed to access such data. Still, our digital archive provides a platform to build from, and we hope that in the future we can include

such data in a similar way as you describe for Antarctica New Zealand.

Thank you for the suggestions regarding the Story Maps. We are glad you found them effective, and will add additional links and resources as future time and funding permits.

---

## Author Response (AR1)

**Authors' Response to Reviews**

Dear editors,

We appreciate the two reviewers' thoughtful comments and detailed assessment of the archive. We have provided our detailed responses to the both reviewers' questions in red, as well as the revised manuscript with associated tracked changes, below.

*Warwick Vincent*

I could not find the supplementary material (Table S1) that is referred to.

Thank you for bringing this to our attention, we have uploaded the supplementary material (Table S1) with the revised manuscript.

Concerning data quality: the data set is accessible via the given identifier, and is complete as a first collection that can now be updated. The data quality is variable – some images are only poorly resolved and have many imperfections (e.g. dust on the slide lens or scanner, poor color rendition), and the metadata are not always complete be-cause the exact date is unknown. The authors might note this by saying that the quality and resolution are variable, but an inclusive approach was taken to maximize the scope of the database.

We agree that it is important to note the variable quality of the data and metadata and have added an explanation to the revised manuscript as suggested.

Further points: The images are taken by many people and "copyright for each item re-mains with the respective contributor" – in that case, how would permission be obtained to reproduce any of these materials, for example in another publication?

Unless otherwise stated, copyright for each item in the database remains with the respective contributor. If a reader wanted to gain permission to use the material in another venue, they would need to contact us or the copyright holder directly. We agree that this is important to note in the manuscript, and have added it to the revised version.

Please give the four bounding coordinates for the geographical area covered in this database.

The bounding coordinates are as follows (decimal degrees): North (-77.504), South (-77.642), East (164.319), West (161.111). These have been included in the revised manuscript.

Please explain the criteria used for selecting the publications for inclusion. They seem incomplete; for example I searched on 'Onyx River' and could only find two publications among

many: 'Spatially discontinuous strain in the "semi-rigid" zone of an ice cliff.1973. 'Principal cation concentrations for a length profile of the Onyx River, Wright Valley. 1973'. It would be useful to include the reports from the Japanese Teams in the 1970s and 1980s (Tetsuya Torii et al., as shown in Fig 1!); these include the classic paper by Yuki Yusa on the solar modeling of Lake Vanda, which is always hard to track down.

Currently, the publications included in the archive are from the three-volume bibliography of Dry Valleys publications referenced in the first paragraph of section 2.1. Together, these three volumes cover published works from 1907 through 1994. We have identified hundreds of additional publications through a Web of Science query covering the entire written record, and plan to add their references to the database in the future. Regarding publications such as those produced by the Japanese research team: these are indeed included in the archive, and the reviewers' comment brought to our attention that the backend of the database had not been configured to index the 'Bibliographic Citation' field, which is why the search was not returning all of the results. We have since adjusted the search configuration, and publications such as those by Yuki Yusa now appear in the search results (e.g., type "Yusa" in the Search bar). The dearth of results when searching for publications on the Onyx River likely reflects the term "Onyx" being seldom used in the title or abstract of earlier Dry Valleys publications. This will change once we include the additional publications identified through the Web of Science.

Line 29 (and twice later) states: "the raw data are available at the Environmental Data Initiative https://doi.org/10.6073/pasta/6744cb28a544fda827805db123d36557"But these are not the raw data – they are metadata (CSV files describing each data item). It seems that the actual data are the photographs, the maps, audio clips etc.

It is correct that the Environmental Data Initiative only holds the metadata. It may be possible to upload the raw data as well in the future, but the reviewer is correct in stating that presently, the raw data are the photographs, maps, etc. available in the digital archive. We thank the reviewer for bringing this to our attention and will clarify this text in the final version of the manuscript.

ESSD Editors may need to comment: this article will have a doi, the collection of meta-data has a doi, but the data itself (the images, maps etc) are in a relational database that does not have a doi assigned - is that OK?

We defer to the editors for this question.

Story Maps: First Western Journey, 1911 Follow the route of the first expedition into the Dry Valleys in 1911. Maybe this should be 'first exploration' because the first expedition visit was in December 1903, led by Scott, who wonderfully proclaimed at that time: "Itis certainly a valley of the dead.

We have amended the text on the story maps page to: "Follow the route of the 1911 expedition into the Dry Valleys, which was the first to spend significant time exploring the region."

*Anonymous Referee #2*

This is just a comment to the authors - I wonder whether the web-browser interface might also be able to house historic US human movement data- similar to a programme of work you probably know called the Antarctic Data Analysis project, run out of Manaaki Whenua - Landcare Research NZ,(https://www.landcareresearch.co.nz/science/soils-and-landscapes/terrestrial-data-analysis-ross-sea-region/data-analysis/human-movement) whereby human movement data (camp and sampling locations) were accessed from Antarctica New Zealand, annual data inputting into a database, and forms a layer in this management tool. Once the tool is up and running, data will be displayed and searchable, to build upa picture of historical human movement (by the NZ programme). It is a combination of these historic photos, maps, oral interviews, publications etc, and data, such as the human movement data, that will be very useful for environmental managers and decision makers as visitor (science and tourist) and environmental pressures increase in the coming years.

We appreciate the comparisons to the Antarctic Data Analysis project from New Zealand, and we think adding annual data on human movement is an excellent suggestion. We have considered this for the archive, but relocating historical movable field camps is very time consuming and we did not have the resources to do so in this project. Moreover, while data on 'modern' (i.e., past 10-20 years) movable camps may exist in some compiled format, the United States National Science Foundation data policy has prevented us from being allowed to access such data. Still, our digital archive provides a platform to build from, and we hope that in the future we can include such data in a similar way as you describe for Antarctica New Zealand.

I liked the story maps. More of these would be great! I am sure the NZ Antarctic Programme has many many photos of the decommissioning of the Vanda Station (https://adam.antarcticanz.govt.nz/nodes/view/37814). It could be great to have all these resources combined and available on one site - but I know this is not possible, but perhaps adding links to other sites with reputable information would be useful.

Thank you for the suggestions regarding the Story Maps. We are glad you found them effective and will add additional links and resources as future time and funding permits.

[revised manuscript text omitted]